# A Workflow for Collecting and Preprocessing Sentinel-1 Images for Time Series Prediction Suitable for Deep Learning Algorithms

**Waytehad Rose Moskolaï** [1,2,*] **, Wahabou Abdou** [3]**, Albert Dipanda** [1] **and Kolyang** [2]

1   Laboratoire Imagerie et Vision Artificielle (ImViA), University of Burgundy Franche-Comté, 21000 Dijon, France
2   Laboratoire de Recherche en Informatique (LaRI), University of Maroua, Maroua P.O. Box 46, Cameroon
3   Laboratoire d'Informatique de Bourgogne (LIB), University of Burgundy Franche-Comté, 21000 Dijon, France
*   Correspondence: waytehad.nkondjock@u-bourgogne.fr; Tel.: +33-755-71-90-04

**Abstract:** The satellite image time series are used for several applications such as predictive analysis. New techniques such as deep learning (DL) algorithms generally require long sequences of data to perform well; however, the complexity of satellite image preprocessing tasks leads to a lack of preprocessed datasets. Moreover, using conventional collection and preprocessing methods is time- and storage-consuming. In this paper, a workflow for collecting, preprocessing, and preparing Sentinel-1 images to use with DL algorithms is proposed. The process mainly consists of using scripts for collecting and preprocessing operations. The goal of this work is not only to provide the community with easily modifiable programs for image collection and batch preprocessing but also to publish a database with prepared images. The experimental results allowed the researchers to build three time series of Sentinel-1 images corresponding to three study areas, namely the Bouba Ndjida National Park, the Dja Biosphere Reserve, and the Wildlife Reserve of Togodo. A total of 628 images were processed using scripts based on the SNAP graph processing tool (GPT). In order to test the effectiveness of the proposed methodology, three DL models were trained with the Bouba Ndjida and Togodo images for the prediction of the next occurrence in a sequence.

**Keywords:** Sentinel-1; C-SAR; satellite images; time series; graph builder; batch preprocessing; prediction; SNAP; dataset; LSTM

## 1. Introduction

In recent years, there have been many technical advances in the remote sensing field. Several thousand Earth observation (EO) satellites are currently orbiting the planet and provide a considerable number of images. With the short revisit time of satellites over an area, it will likely soon be possible to get free daily images of any surface on the world [1]. Since 2014, for instance, the Copernicus program, launched by the European Union and the European Space Agency (ESA), has provided data from a constellation of six satellites called the Sentinel mission (Sentinel-1 to Sentinel-6). The first mission of this series, Sentinel-1 (S1), operates in active mode and supplies images every six days, regardless of the weather or the time of day (e.g., night, cloud cover, fog). Compared to optical Sentinel-2 (S2) images, S1 radar images have the advantage of being cloud-free. This quality allows them the ability to perform better in many applications; however, before use, raw satellite data must be preprocessed. In practice, the processing operations for S1 images are more complex and diverse than those of S2 images, which often limits their uses.

Consecutive data of an area, acquired from several dates, are called satellite image time series (SITS) [2–5]. Satellite image time series applications are varied and include agricultural resources monitoring, environment management, forest mapping, anomaly detection, and SITS prediction [6,7]. For the case of SITS prediction tasks, deep learning (DL) techniques are more and more mentioned in the literature. In particular, long-short-term

memory (LSTM) architectures such as ConvLSTM, CNN-LSTM, and Stack-LSTM are used for their ability to deal with image sequences [8].

LSTM networks are a particular type of recurrent neural network (RNN) introduced by Hochreiter and Schmidhuber in [9]. The LSTM architectures are generally used with data organized in sequences and have cells that store the state from previous layers [2].

DL algorithms for time series prediction typically require datasets made of long image sequences [10–12]; however, the complexity of SITS preprocessing tasks and the lack of training datasets are some limitations related to the use of DL techniques, as mentioned in [2]. Indeed, for works on the prediction of land cover classes using DL algorithms, for example, it is not easy to find datasets with SITS already preprocessed. Starting from scratch to build a long series can also be difficult for non-experts in the remote sensing domain. To resolve this, it has been suggested to reveal methods for automatic image batch processing and increase the number of available training datasets. In this way, the number of works using DL algorithms for SITS prediction could considerably increase. An interesting application in this field is for example the prediction of deforestation from SITS with DL approaches.

There are several publicly available remote sensing databases to use with DL algorithms [13–16]; however, these datasets are mostly used for image classification tasks [17,18]. For example, the authors in [16] proposed a georeferenced dataset for training and validation of deep learning algorithms for flood detection for Sentinel-1 images. In [13], an image dataset for ecological investigations of birds in wind farms was constructed. The authors in [14] also proposed OpenSARUrban, a Sentinel-1 dataset dedicated to the content-related interpretation of urban SAR images.

Moreover, although some studies have explored the problem of a lack of datasets [19–25], there are few studies that have addressed the constitution of datasets suitable for SITS forecasting problems. For ship detection, for example, the authors in [25] carried out a study in which ship positions have been automatically extracted in batch from Sentinel-1A images, and in [15], the Large-Scale SAR Ship Detection Dataset-v1.0 from Sentinel-1 have been proposed. In [20], a workflow allowing the production of a set of preprocessed Sentinel-1 GRD data was presented. However, the proposed scripts allow the preprocessing of the images one by one.

Within this scope, the goal of this paper is to propose a complete workflow with scripts for collecting and batch processing S1 images. Then, a method to prepare the datasets to use with the DL algorithms is presented. The purpose of this experiment is to build public datasets of SITS that can be used for multiple other purposes. The code used to collect images in bulk was generated from NASA's Vertex platform. For preprocessing, the scripts were based on the graph processing tool (GPT), proposed by the engine part of the Sentinel application platform (SNAP) software, Version 8.0 [26]. This special command based on the SNAP graph processing framework (GPF) is used to process raster data in batches, through extensible markup language (XML) files containing graph operators.

To test the effectiveness of the proposed methodology and demonstrate the practicality of the built datasets, the CNN-LSTM, ConvLSTM, and Stack-LSTM architectures were used to design DL models for the next frame forecasting in S1 image sequences. Experiments have shown that longer time series provide better performance than shorter ones.

Compared to similar works in the literature, the methodology proposed in this study has the following particularities: (1) the process described in this paper allows us to easily build datasets of SITS suitable for prediction tasks with DL algorithms. The images of each series allow us to represent the different states of the land cover classes thanks to the data acquired over time. Contrary to the dataset used mainly for classification tasks, the data collected here are more adequate for forecasting problems; (2) the procedure is complete, going from data collection to data preparation; (3) the proposed preprocessing chains allow us to execute operations in batch and not individually; (4) the VV and VH bands of images are separated during the process to allow a better analysis; (5) the preprocessed datasets and all the scripts used in this study are available to users.

The experiments conducted on three study areas, namely the Bouba Ndjida National Park (BNP), the Dja Biosphere Reserve (DBR), and the Wildlife Reserve of Togodo (WRT) provided the preprocessed datasets for time series applications. The land cover classes on the selected AOI are diverse. This will allow us to create several types of DL models for many applications.

In sum, the contributions of this paper are as follows:

- Proposal of a complete workflow for S1 image time series collection and preprocessing;
- Development of a GPT-based script for S1 images batch processing that can be easily modified by users;
- Construction of a public database made of three SITS;
- Presentation of a method for preparing image time series to use with DL algorithms.

The rest of this paper is organized as follows: Section 2 describes the data used for this study. Section 3 presents the methodology and the tools used in this paper. Section 4 depicts the results obtained from the study areas. Finally, Section 5 concludes the paper.

## 2. Used Data

### 2.1. Main Characteristics

The S1 satellites are the first in the family of six EO satellite missions (Sentinel-1 to Sentinel-6). The S1 mission consists of a constellation of two satellites namely Sentinel-1A (S1A), launched on 3 April 2014, and Sentinel-1B (S1B) launched on 25 April 2016. The two satellites S1A and S1B share the same orbital plane. Together they provide images every six days, day and night, regardless of the weather conditions. Each of the two satellites flies over the same area every 12 days [8].

The S1 sensors are equipped with synthetic aperture radar (SAR)-type instruments that operate in the C-band. These sensors, called C-SAR, can acquire data covering large areas (up to 400 km) and can reach five meters resolution in four exclusive modes: stripmap mode (SM), interferometric wide swath (IW) mode, extra wide swath (EW) mode and wave mode (WV). Thanks to a transmission chain and two parallel reception chains for horizontal (H) and vertical (V) polarizations, the S1 C-band SAR instruments operate in single polarization (HH or VV) and dual polarization (HH + HV or VV + VH). The products acquired in SM, IW, and EW modes are available in single or dual polarization while the products acquired in WV are only available in single polarization.

The Sentinel images are free of charge to all users, including the general public and commercial users. All the products are distributed in the Sentinel standard archive format for Europe (SAFE). For each mode, images are available in three levels corresponding to the processing level: Level-0, Level-1, and Level-2. The Level-0 products correspond to unfocused raw SAR data. These data must be decompressed and preprocessed in order to be usable. The Level 1 products are obtained by processing the Level-0 products. Level-1 products are the most used in common applications. They are the base data from which Level-2 products are derived and feature a wide swath (250 km) with high resolutions (geometric and radiometric). Sentinel-1 mission provides data for applications in the Copernicus priority areas such as maritime monitoring, emergency management, and land monitoring. In the field of land monitoring, S1 images are mainly used for forestry (sustainable forest management, forest type classification, biomass estimation, and disturbance detection), agriculture (monitoring of crop conditions, soil properties, and mapping tillage activities), and urban deformation mapping (monitoring of land subsidence, structural damage, and underground construction).

In this study, Level-1 ground range detected (GRD) data acquired in IW mode and in dual polarization (VV + VH) were used. The S1-GRD products do not retain phase information, but only amplitudes. The resulting images have pixels with approximately square size (10 m $\times$ 10 m).

*2.2. Study Areas*

The satellite images are widely used to assess the effectiveness of protected areas [27]. The images acquired and processed in this study are S1A data corresponding to three distinct protected areas: the Bouba Ndjida National Park (BNP), the Dja Biosphere Reserve (DBR), and the Wildlife Reserve of Togodo (WRT).

2.2.1. Bouba Ndjida National Park

The BNP is one of the national parks in Cameroon, a Central African country. It was established in 1968 and is located in the northern region, bordering Chad. As shown in Figure 1, this area is crossed by numerous rivers. Geographically, the BNP is located between 08°21′ and 09° north latitude, and between 14°25′ and 14°55′ west longitude [28]. Although it is the largest national park in the country (with a total area of approximately 220,000 ha), it is also the most isolated. Its landscape is subject to strong pressure from human activities and natural phenomena. Regarding the vegetation, it is mainly made up of Sudano–Guinean shrub savanna. The climate is Sudano–Sahelian with a long dry season (8 months) and a short rainy season (4 months). Since August 2011, a partnership agreement has been signed between Chad and Cameroon to create the BSB (Binationale Séna Oura–Bouba Ndjida) complex in Yamoussa [29].

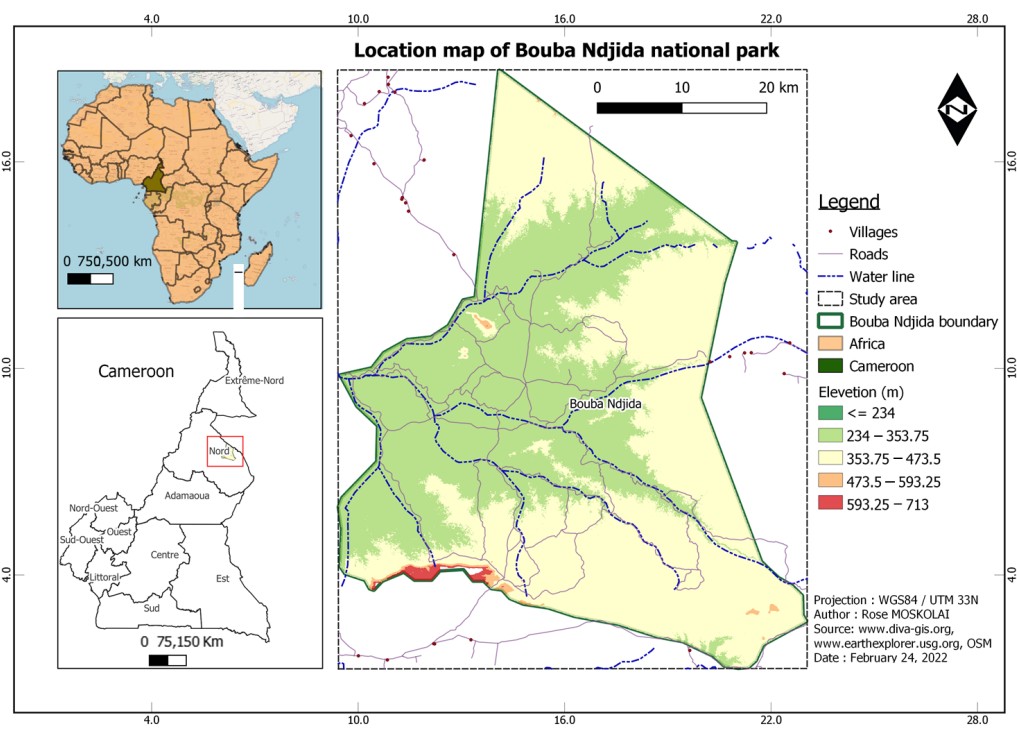

**Figure 1.** Location map of the Bouba Ndjida National Park.

2.2.2. Dja Biosphere Reserve

The DBR, which takes its name from the Dja River that surrounds it, is located between 2°40′ and 3°23′ north latitude and 12°25′ and 13°35′ east longitude. The DBR covers 526,000 ha and is one of the largest protected areas of the Guinea–Congolian tropical rain forests [30]. As shown in Figure 2, this reserve is located in the south of Cameroon. The terrain is fairly flat and the landscape is composed of a multitude of small hills; however, in the south part, the topography is more rugged with cliffs, rapids, and waterfalls. According to the 2005 Cameroonian population census, the number of people living in the neighboring localities was 26,153 [31]. There are four seasons in the DBR area: a long rainy season, a long dry season, a short rainy season, and a short dry season [31].

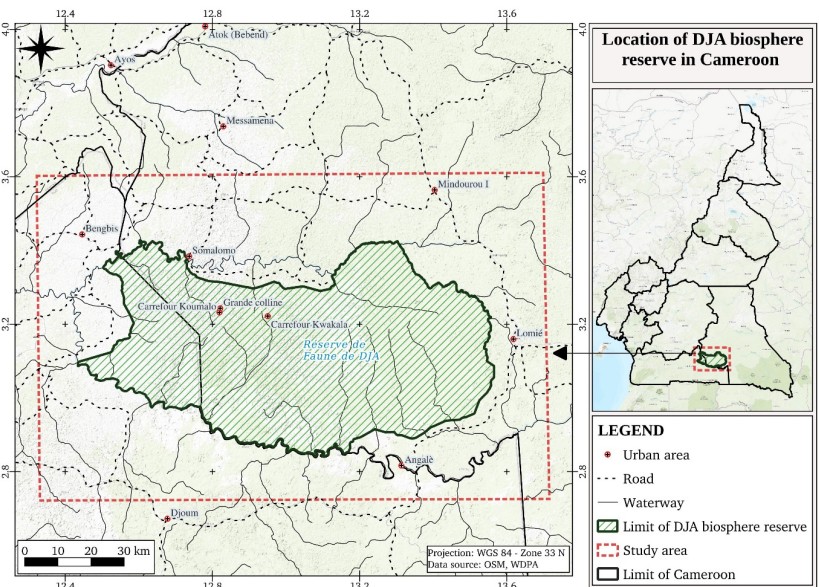

**Figure 2.** Location map of the Dja Biosphere Reserve.

### 2.2.3. Wildlife Reserve of Togodo

The WRT is located in the southeast of Togo, a West African country. It is located between 6°23′ and 7° north latitude and between 1°23′ and 1°34 west longitude [32]. With an estimated area of 30,000 ha, the WRT is bounded by the Mono River in the east and by the Akpaka and Afan Rivers, as well as the Kpové and Tsafé farms in the west [33]. The human occupation around the WRT includes various ethnic groups that have been settled in the area for more than three centuries [34]. The fourth general census of population and housing of 2010 in Togo estimates the population density between 100 and 150 inhabitants/km². The vegetation in WRT includes dense forest; gallery forest; open forest; wooded, shrubby, and grassy savannah; cultivated areas; fallow land. The hydrophilic vegetation is also very rich in flowers, with about 210 species. In addition, palm oil plantations are very developed in this area and are considered a source of deforestation. Figure 3 shows the location map of the WRT.

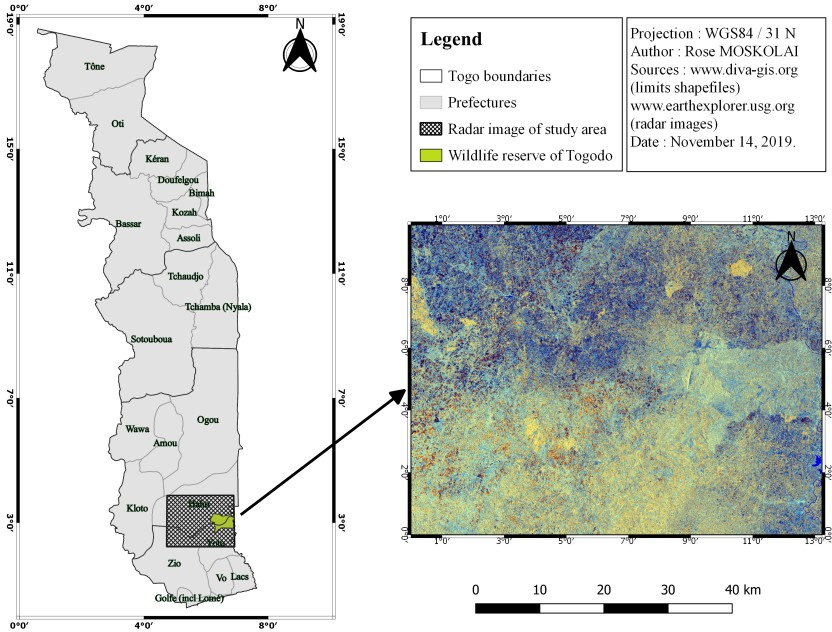

**Figure 3.** Location map of the Wildlife Reserve of Togodo.

### 3. Methodology

The estimation of future values in a time series is always based on the observation of previous values. The time series prediction task is more difficult than classification and regression problems because of the complexity added by the order and temporal dependence between the data.

Several prediction methods exist and allow us to obtain very good results for some problems. For linear time series, the main methods are regression models (such as the auto-regressive integrated moving average), exponential smoothing models, and the other linear models that have been very popular for the last thirty years [35]. However, these classical methods have some limitations. For example, they require the use of complete sets (missing or corrupted data usually degrade the performance of the model). These methods are also suitable for linear series or series with a constant time step, focus on one-step prediction problems, and concentrate on the use of uni-variate data [6].

For problems such as the prediction of land use and land cover changes using satellite imagery, classical models are not very suitable because SITS are non-linear and sometimes, there are missing or corrupted data in the series. CA-Markov methods are widely used in the literature for this kind of problem, but these methods require an important step of features engineering and the use of auxiliary data which strongly influence the prediction results [36]. Moreover, DL models can automatically learn complex mappings between inputs and outputs. Feature extraction is a very laborious and time-consuming step; this step is done automatically with DL algorithms, unlike classical machine learning methods where the users have to be extremely accurate in the supervised learning process. The authors in [37] have compared classical methods and DL algorithms for time series prediction problems and have shown for example that DL methods are effective and easier to apply than classical methods.

The powerful deep learning capabilities hold great promise for time series forecasting, especially for problems with complex and non-linear dependencies such as in SITS. The abilities of neural networks such as multi-layer perceptrons (MLP), CNN, or LSTM networks allow reaching performances that surpass those of classical methods: (1) neural networks are robust to noise in the data and can produce good results even with non-linear series or with missing or corrupted data; (2) the capabilities of CNNs to automatically extract important features from the raw input data can be applied to time series prediction problems; (3) LSTM networks can store temporal dependencies between data [38–40].

So, for complex tasks such as SITS forecasting, the capabilities of DL models seem well suited, provided that a dataset with a long time series is available. In fact, the volume of historical available data representing the same area is important for the DL model to better capture the relationships between past and present images. Some events may occur, for example, only after a long period of time; it is important for the model to have a long enough series to capture the behavior of that event over time. The longer the series, the more training data the model can access; this leads to greater prediction accuracy. Having a long time series also avoids the phenomenon of overfitting that occurs in DL models when it focuses on the training data and fails to make good predictions on new data [41].

Since collecting and preprocessing satellite images is not always a straightforward task, we present in the following subsections methods to facilitate batch collection and preprocessing of Sentinel-1 image time series that are suitable for time series prediction with DL algorithms. In addition, the process of preparing preprocessed data for use with DL models is also described.

#### 3.1. Bulk Collection

All types of S1A images can be downloaded for free from several platforms. Each platform has its own particularities. The most widely used platforms are as follows:

- The Copernicus Open Access Center (https://scihub.copernicus.eu/ (accessed on 10 May 2022)) provides users with full access to Sentinel products. Images are available online as soon as satellite images are received; however, older images are generally not

immediately downloadable and must be specifically requested. About six months after the registration of an image, the image is put in offline mode on the official Copernicus website, and a request must be sent to the server to put the product back online, for a determined period.

- The Copernicus Dedicated Access Center (https://cophub.copernicus.eu/ (accessed on 10 May 2022)) is for project services such as the provision of image archives that are no longer available on the main server.
- The NASA Vertex platform or Alaska Satellite Facility (ASF) (https://search.asf.alaska.edu/#/ (accessed on 10 May 2022)) provides the whole image time series of an area online. In other words, there are no offline products in this data center. Another advantage of this platform is the fact that the area of interest (AOI) can be imported as a shapefile or in well-known text format (WKT).
- The Copernicus Access Center mirrors https://peps.cnes.fr (accessed on 10 May 2022) (PEPS) and https://code-de.org (accessed on 10 May 2022) (code-de) proposed by the National Center of Space Studies, known as Centre d'Etude Spatiale (CNES), and the German Space Agency, respectively.
- The EO platform (EO Browser) (https://apps.sentinel-hub.com/eo-browser/ (accessed on 10 May 2022)) allows users to browse and compare images from all the provided data collections.

In most platforms, images are generally downloaded individually; however, when a series of images must be collected, it is preferable to automate the downloading of the data using scripts, and, one way to do this is using dhusget program, a download command line based on the cURL and Wget programs and initiated by the Sentinel Data Center. The use of dhusget is completed by a command line query, which has the following structure:

```
dhusget.sh [LOGIN]  [SEARCH_QUERY]  [SEARCH_RESULTS]  [DOWNLOAD_OPTIONS]
```

More details on the various setting options are available on the Copernicus Data Access Center user page. (Online (Available): https://scihub.copernicus.eu/userguide/BatchScripting (accessed on 10 May 2022).)

Another simple way to automatically download S1 images is using aria2c, which is the download manager on the official Copernicus website. With this tool, the user specifies the search criteria related to the AOI, and the validation of the selected products generates a file named `products.meta4`. The command used to start the batch download is listed below. Note that a valid Copernicus Open Access Center user account is required to run these command lines.

```
aria2c  --http-user='username' --http-passwd='password'
        --check-certificate=false
        --max-concurrent-downloads=2
        -M products.meta4
```

A major drawback with the two methods is when a product is offline, the command is not able to immediately download it; however, there is a way to configure the script to send a request to the server when a product is offline, and check its availability after a certain time.

For the data used in this study, the researchers downloaded images from the ASF platform and selected all the available images for each study area. The corresponding Python scripts were executed in command lines on a local machine to start the recording of all images. For more information about the bulk collection from the Vertex platform, please navigate to the ASF website. (Online (Available): https://asf.alaska.edu/how-to/data-tools/data-tools/ (accessed on 3 July 2022).)

The files containing the source code used to download the images from the three study areas are available at http://w-abdou.fr/sits/ (accessed on 10 August 2022).

Table 1 presents the search parameters used for each study area and all available data from 15 June 2014 to 3 July 2022 were selected to be downloaded.

**Table 1.** Search options.

| Site | Mission | File Type | Beam Mode | Direction | Area of Interest (WKT) |
|------|---------|-----------|-----------|-----------|------------------------|
| BNP | S-1A | Level-1-GRD | IW | ASCENDANT | POLYGON (14.6088 8.7979, 14.6793 8.7979, 14.6793 8.824, 14.6088 8.824, 14.6088 8.7979) |
| DBR | S-1A | Level-1-GRD | IW | ASCENDANT | POLYGON (12.1466 2.9639, 12.8304 2.9639, 12.8304 3.2727, 12.1466 3.2727, 12.1466 2.9639) |
| WRT | S-1A | Level-1-GRD | IW | ASCENDANT | POLYGON (1.398 6.761, 1.611 6.759, 1.613 6.930, 1.400 6.932, 1.398 6.761) |

The AOI to be downloaded are delimited using the WKT coordinate; thus, to download the images corresponding to a new location, one needs to know the geographical coordinates and enter them on the platform to generate a corresponding Python file. The command to execute the download file (on a Unix console) is the following:

```
Python name_of_python_file.py
```

*3.2. Preprocessing*

3.2.1. SNAP Software

For the processing of Sentinel data, the ESA has developed a powerful application called the Sentinel Application Platform (SNAP). The SNAP software was jointly developed by Brockmann Consult, SkyWatch, and C-S for the visualization, processing, and analysis of EO data in general and Sentinel products in particular. As Figure 4 displays, SNAP uses several technologies such as the Geospatial Data Abstraction Library, NetBeans Platform, Install4J, GeoTolls, and Java Advanced Imaging (JAI). The processing can be done using the SNAP Desktop graphical interface or command line in the SNAP Engine.

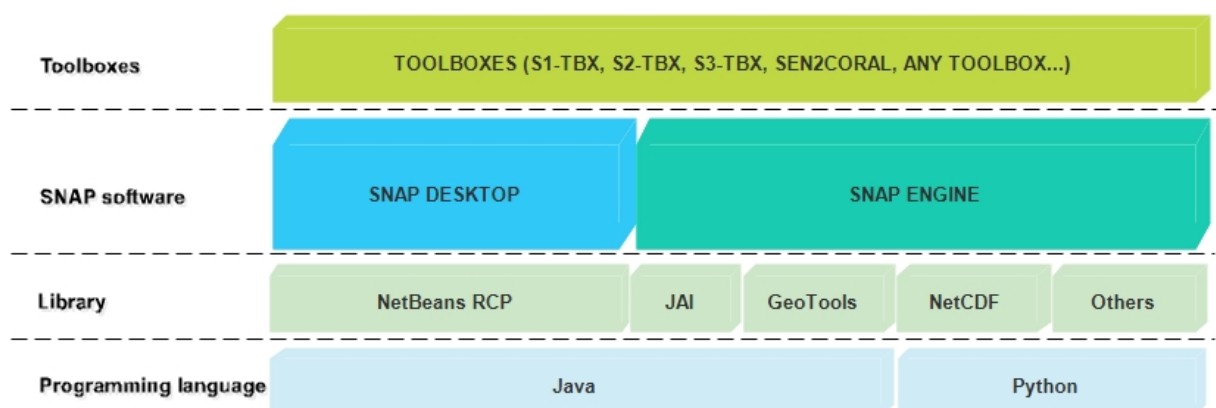

**Figure 4.** General architecture of SNAP software operations.

Within the SNAP software, there is a flexible processing framework called GPF [26]. The GPF is based on JAI and allows users to implement custom batch processing chains. Figure 5 highlights how the GPF works using assembled graphs from a list of available operators. It is a directed acyclic graph (without loops or cycles) where nodes represent the processing steps called operators, and edges indicate the direction in which data is transferred between the nodes. The data source is the images received as inputs from the read operator, and the output can be either an image recorded by the write operator or a displayed image. Each data passes through an operator, which transforms the image, and the resulting image is sent to the next node until it reaches the output. No intermediate files are recorded during the process unless a write operation has been intentionally introduced into the graph.

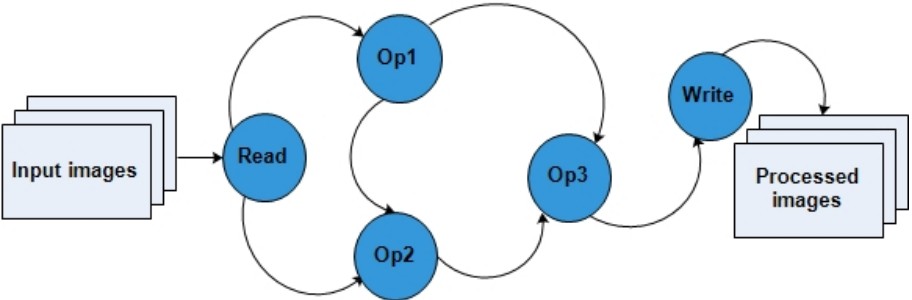

**Figure 5.** Data processing in the SNAP GPF tool. Op1, Op2, and Op3 represent the operators.

In sum, the GPF architecture offers the following advantages:

- No overloading of input/output operations;
- Efficient use of storage memory (no writing of intermediate files);
- Re-usability of the processing chains;
- Ability to reuse operator configurations;
- Parallel processing of graphs according to the number of available cores.

### 3.2.2. Preprocessing Workflow

Prior to the development of the batch script, a processing workflow was built using the SNAP graph builder tool. The designed graph representing the operators applied to the study's data is schematized in Figure 6.

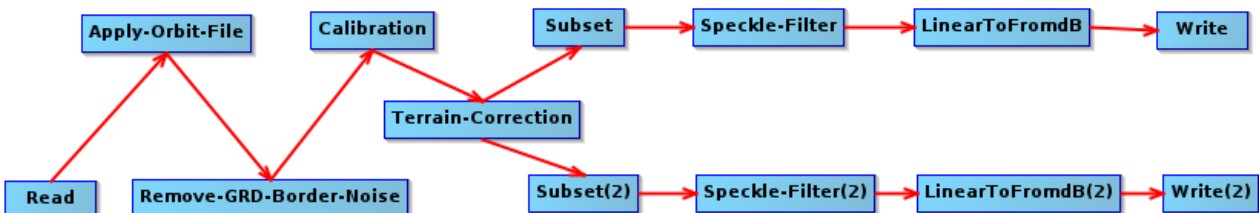

**Figure 6.** The proposed SNAP graph corresponding to Sentinel-1 images preprocessing workflow.

1.  Read: The "read" operator is used to load the data. This operator accepts not only S1 images but also other types of data.
2.  Apply orbit file (AOF): The default metadata provided when SAR data are downloaded is generally not very accurate. In SNAP, the updated orbit file, with more accurate information to help improve geocoding and other processing results, is automatically downloaded by the AOF operator. This operation should be performed as a priority before all other preprocessing steps for better results.
3.  Border noise removal (BNR): Because of irregularities on the Earth's surface, deformations can appear on the Level-1 images during their generation. The BNR operation allows one to correct and remove noises present on the edges of images.
4.  Calibration: The calibration step (calibrate) aims to correct the signal intensity according to the sensor characteristics and the local incidence angle. The metadata of the input products allows SNAP to automatically determine the corrections to be applied. In this work, the operation is performed for both VV and VH polarizations.
5.  Terrain correction: The initially downloaded S1 images are devoid of geographic coordinates, which makes them unusable for most applications. The ortho-rectification step is performed to geo-reference images in order to project them into the Universal Transverse Mercator system or World Geodetic System 1984. This process also allows one to correct the distortion effects that occurred during the acquisition (overlay, shading). The terrain correction operation is based on a digital elevation model (DEM) automatically downloaded by SNAP or provided by the user.

6.  Image clipping (subset): Each downloaded S1 image covers an area of 250 km × 250 km (swath). For smaller AIO, it is often not necessary to use the whole image but to extract a region. This can be done in SNAP using the subset operator, which slices the image according to a specific area. Moreover, cutting an image before performing other processing reduces memory consumption, processing time, and storage. The geographic coordinates of the AOI in WKT format may be required for this step.

7.  Filtering: Radar images contain some noise called speckle, which degrades them and makes them unusable in most cases. To improve the quality of the data and their analysis, it is useful, in some cases, to reduce the speckle effect using filtering. The speckle filtering operation allows one to remove as much noise as possible and improve the readability of the images. There are several filtering algorithms and methods such as simple spatial filtering, spatio-temporal filtering, or multi-temporal filtering.

8.  Converting values to decibels: Before writing the data to disk, the last step in the processing chain is the conversion to decibel scale to convert the initial pixel values to decibels (dB). In fact, SAR imagery has a high value range, and the decibel transformation is made to improve data visualization and analysis. Typically, this is done using a logarithmic transformation that stretches the radar backscatter over a more usable range that has a nearly Gaussian distribution.

9.  Write: The last operation of the process is the writing of the resulting data to the disk.

Each of the used operators are presented below and the corresponding parameters are described in Table 2.

**Table 2.** The parameters used for each operator.

| Operator | Parameter | Value |
|---|---|---|
| Read | Data format | Any format |
| Apply orbit file | Orbit state vectors<br>Polynomial degree | Sentinel precise (Auto download)<br>3 |
| Border noise removal | Polarizations<br>Threshold<br>Polarizations | 500<br>0.5<br>VH, VV |
| Calibration | Polarizations<br>Output sigma0 band | VH, VV<br>True |
| Terrain correction | Source bands band<br>DEM<br>DEM resampling method<br>Image resampling method<br>Pixel spacing<br>Map projection<br>Selected source bands | Sigma0_VH, Sigma0_VV<br>SRTM 3 s (Auto download)<br>Bilinear interpolation<br>Bilinear interpolation<br>10 m<br>MP1 [1] for BNP and DBR; MP2 [2] for WRT<br>True |
| Subset | Source bands<br>Copy metadata<br>Geographic coordinates | Sigma0_VH<br>True<br>WKT1 [3] (BNP), WKT2 [4] (DBR) and WKT3 [5] (WRT) |
| Subset (2) | Source bands<br>Copy metadata<br>Geographic coordinates | Sigma0_VV<br>True<br>WKT1 [3] (BNP), WKT2 [4] (DBR) and WKT3 [5] (WRT) |
| Speckle filtering | Source bands<br>Filter | VH<br>Refined Lee |
| Speckle filtering (2) | Source bands<br>Filter | VV<br>Refined Lee |
| Linear to dB<br>Linear to dB (2) | Source bands<br>Source bands | VH<br>VV |
| Write | Save as | BEAM-DIMAP |
| Write (2) | Save as | BEAM-DIMAP |

[1] MP1 = EPSG:32633-WGS84/UTM 33N; [2] MP2 = EPSG:32631-WGS84/UTM 31N; [3] WKT1 = POLYGON ((14.411 8.480, 14.914 8.480, 14.914 8.996, 14.411 8.996, 14.411 8.480, 14.411 8.480)); [4] WKT2 = POLYGON ((12.480 2.904, 12.480 3.350, 13.369 3.350, 13.369 2.904, 12.480 2.904)); [5] WKT3 = POLYGON ((1.398 6.761,1.611 6.759,1.613 6.930,1.400 6.932,1.398 6.761)).

3.2.3. Batch Processing Using GPT

Each graph designed with the SNAP graph builder can be exported as an XML file to automate the process. Even though it is possible to run batch processing from SNAP software, it is recommended to execute operators via the SNAP GPT tool for greater flexibility. The GPT tool is the command-line interface of SNAP that is used to execute raster data operators in batch mode. This is another way to perform batch processing without using the graphical user interface mode within the SNAP software. To use the gpt command, the operations can be processed individually, or they can be processed in a defined workflow.

Instead of specifying operator parameters separately in the processing stream, it is possible to use the XML-encoded file as a parameter of the gpt command. The general syntax used to execute the command for one image is offered below.

```
gpt <GraphFile.xml> [ options ] [<sourcefile1> <sourcefile2> ...]
```

All the XML files used to process images corresponding to the three study areas have almost the same content. The main difference lies in the geographical coordinates used to clip the images (subset operator) and the map projection system (terrain correction operator).

To simultaneously execute the processing of all the raw images, the researchers proposed a bash script. The code lines were written to be executed in a Unix environment. This script browses the whole directory and applies the processing chain to each found image. The advantage of having each file name written according to a particular naming convention (Online (Available): https://sentinels.copernicus.eu/web/sentinel/user-guides/sentinel-1-sar/ (accessed on 10 May 2022)) is that it allows one to receive the acquisition date of each image and rename the processed image. For instance, a raw image name is renamed after processing into 20150409_VH.dim for the VH band and 20150409_VV.dim for the VV band. This change facilitates the subsequent use and manipulation of the processed images.

```
S1A_IW_GRDH_1SDV_20150409T171237_20150409T171302_005410_006E19_7760.zip
```

The XML files defining all the preprocessing operations and the scripts to execute data in the batch are available at http://w-abdou.fr/sits/ (accessed on 10 August 2022).

At this stage of the preprocessing workflow, the images can be used for many other applications such as classification, segmentation, spatio-temporal analysis, and so on. For each use case, specific operations must be performed on the preprocessed data. For the specific case of using datasets for SITS prediction, the operations described in the following section prepare the data so that it can be provided to DL networks.

3.2.4. Data Preparation for Use with DL Algorithms

Once the SITS are available, the next step is to prepare the data for use with DL algorithms. Indeed, time series data must be transformed before it can be used in a supervised learning model.

In a univariate supervised learning problem, there are input variables $(X)$, output variables $(Y)$, and a model that uses an algorithm to learn the mapping function from the input to the output: $Y = f(X)$ [8]. In this paper, the researchers present a method used to transform univariate time series (univariate time series are sequences of data consisting of a single set of observations with a temporal order (such as SITS)) to solve sequence-to-one forecasting problems where the inputs are a sequence of data and only one occurrence of data is offered as an output.

First, to simplify the use of the DL algorithms, all the images are transformed into JPG files, normalized, and reshaped to smaller sizes before the preparation step. Each image of the series has the shape $(W, H, N)$, where $W$, $H$, and $N$ denote the number of rows, columns, and channels for each image, respectively. Note that chronological order of images must be kept in each data since the researchers were studying forecasting tasks, and 80% of the data was selected for model training with the remaining 20% for testing.

Next, the DL model to design requires that data are provided as a collection of samples, where each sample has an input component ($X$) and an output component ($Y$), as presented in Table 3. This transformation allows one to know what the model will learn and how the model can be used to make predictions. After transforming the data into a suitable form, they are represented as rows and columns.

**Table 3.** A univariate time series converted to supervised learning.

| X_Train | Y_Train |
|---|---|
| $[X_1, X_2, X_3, X_4, X_5]$ | $X_6$ |
| $[X_2, X_3, X_4, X_5, X_6]$ | $X_7$ |
| $[X_3, X_4, X_5, X_6, X_7]$ | $X_8$ |
| ... | ... |
| $[X_{t-5}, X_{t-4}, ..., X_{t-1}]$ | $X_t$ |

For two-dimensional data using DL networks, such as convolutional neural networks (CNN) or long-short term memory (LSTM) for prediction, additional transformations are required to prepare the data before fitting models. Thus, the data are transformed to the form of ($samples, timestep, W, H, features$), where $samples$ are the sequences, $timestep$ is the number of occurrences in each sample, and $features$ correspond to the number of variables to predict.

In sum, to prepare the time series data for fitting predictive DL models, the following actions are necessary:

- Resize the data;
- Organize data into a training set and test set (80% and 20%);
- Split the univariate sequence into samples (generate $X\_train$ and $Y\_train$ samples);
- Reshape data from $[samples, timestep]$ into $[samples, timestep, W, H, features]$.

## 4. Results

### 4.1. Used Tools

The experiments were carried out on a computer with a GNU/Linux operating system (Ubuntu 20.04.1 LTS). Table 4 gives the main hardware characteristics of the system used in this study.

**Table 4.** Hardware characteristics. CPU: central processor unit.

| | |
|---|---|
| Memory | 96 gigabytes (GB) |
| Processor | Intel Xeon (R) CPU E5-2609 v4 |
| Processor frequency | 1.70 GHz × 16 |
| Graphic card | NVD9 |
| Hard Disk | 1.5 terabyte (TB) |

As far as software tools are concerned, the following programs were used:

- SNAP 8.0: A tool used to design and implement the automatic batch processing workflow. This software also allowed to display and analyze the downloaded data;
- QGIS 3.16.0-Hannover: A program used for manipulating the shape-files and the geographic information. This tool was also used to analyze the obtained image time series;
- Python 3: A programming language mainly used for download scripts;
- Tensorflow: A Google open source tool dedicated to machine learning;
- Google Colaboratory environment (Colab pro) (www.colab.research.google.com (accessed on 10 August 2022)): to have a privileged access cloud to a graphical processor unit (GPU).

### 4.2. Preprocessed Dataset Description

At the end of this study, a database of three time series of preprocessed S1 images is built. The SITS are from the selected study areas, namely the BNP, the DBR, and the WRT.

During the preprocessing chain execution, no intermediate files are saved. In fact, only the final results are saved on the disk. Since the sizes of the raw images are very large, this avoids unnecessary use of storage memory.

A total of 628 images are downloaded with the Python programs. Each raw image, with an average size of 1 GB, is downloaded as a compressed file (zip) using an average rate equal to ten megabytes per second (MB/s). It took about 18 h to download all the data with a total size of 654.9 GB. One can notice that a fast internet connection is recommended for this step. The summary of the download information is given in Table 5.

**Table 5.** Download summary.

| Site | No of Files | Total Size | Total Time | Average Rate |
|------|-------------|------------|------------|--------------|
| BNP | 204 | 198.1 GB | 6.2 h | 9.94 MB/s |
| DBR | 207 | 247.0 GB | 6.2 h | 10.01 MB/s |
| WRT | 217 | 209.8 GB | 5.6 h | 10.04 MB/s |

The analysis of the collected images depicts that some data are missing in the time series. Indeed, there is about 6% of missing data for the BNP and DBR areas, compared to only 2% for the WRT. Table 6 gives the list of dates (dd/mm/yyyy) where images are not available for each study area. The column "Percentage" represents the proportion of missing data.

**Table 6.** List of dates corresponding to the missing data.

| Study Area | No of Missing Data | Percentage | Dates |
|------------|--------------------|------------|-------|
| PNB | 15 | 6.85% | 21/04/2015, 03/05/2015, 15/05/2015, 08/06/2015, 20/06/2015, 07/08/2015, 31/08/2015, 24/09/2015, 05/10/2015, 17/10/2015, 29/10/2015, 27/02/2016, 27/04/2016, 26/06/2016, 29/03/2017, 21/06/2017, 20/12/2019. |
| DBR | 14 | 6.33% | 19/04/2015, 01/05/2015, 13/05/2015, 30/06/2015, 29/08/2015, 10/11/2015, 27/12/2015, 01/02/2016, 25/02/2016, 25/04/2016, 24/06/2016, 12/12/2020, 12/01/2021, 12/05/2022. |
| WRT | 05 | 2.25% | 02/05/2015, 29/10/2015, 28/12/2015, 25/06/2016, 22/05/2018. |

The raw satellite images contain a lot of information and are very large. Their processing is often time-consuming and requires a processing unit with good capabilities. Table 7 presents the performed processing time for each study area. In sum, about 16 h were required for the three areas. The WRT presents the smallest processing time because the surface considered for the raw image slicing was the smallest, only 44,379 hectares (ha).

**Table 7.** Processing time for each study area.

| Site | Processing Time | Subset Surface |
|------|-----------------|----------------|
| BNP | 4 h and 45 min | 220,667 ha |
| DBR | 8 h and 4 min | 486,964 ha |
| WRT | 3 h 48 min | 44,379 ha |

The constructed datasets (Figure 7) are available at http://w-abdou.fr/sits/ (accessed on 10 August 2022) and can be downloaded for free. A total of 1256 preprocessed images in VV and VH polarization are proposed: 408 for BNP, 414 for DBR, and 434 for WRT.

Table 8 gives the characteristics of each dataset. Since the pixel size of the images is 10 m, the areas in hectares for each of the AOI are approximately 221,000 for BNP, 490,000 for DBR, and 44,500 for WRT.

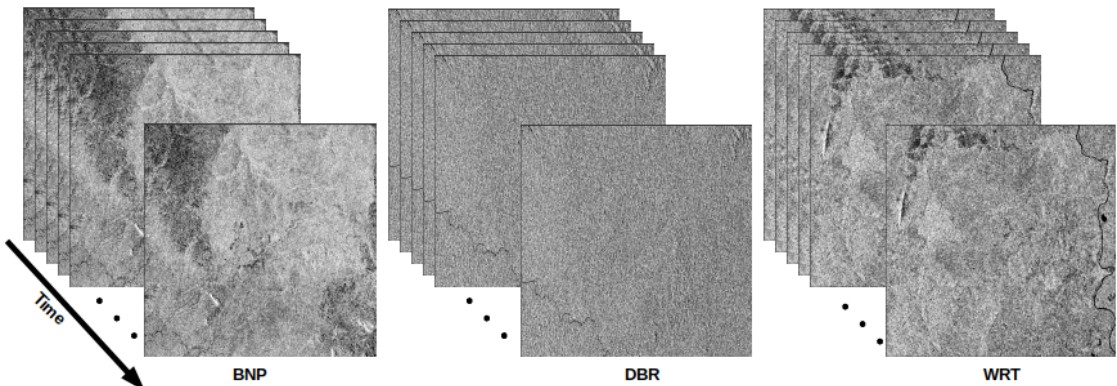

**Figure 7.** Representation of VH-bands for BNP, DBR, and WRT S1 image time series.

**Table 8.** Main characteristics of the preprocessed datasets.

| Site | Image Size | Pixel Size | Dates | No of Images | Total Size | Data Type | File Type |
|------|-----------|-----------|-------|-------------|-----------|-----------|-----------|
| BNP | 5553 × 3980 | 10 m | From 09 April 2015 to 01 July 2022 | 408 | 39.2 GB | Float32 | DIMAP |
| DBR | 9899 × 4948 | 10 m | From 07 April 2015 to 29 June 2022 | 414 | 84.7 GB | Float32 | DIMAP |
| WRT | 2380 × 1869 | 10 m | From 27 March 2015 to 30 June 2022 | 434 | 11.1 GB | Float32 | DIMAP |

Figure 8 shows plots extracted from each AOI. It allows distinguishing globally the type of land cover in each image, thanks to Figure 8a, which corresponds to the RGB image from Google satellite images. Google satellite images show a natural view of surfaces and are imported from the QGIS Tile+ plugin. For example, one can deduce from Figure 8a that the DBR area is essentially made up of forest cover, while in the BNP there is a lot of savannah or bare soil. The red rectangle shown in Figure 8c delineates the area extracted from the base image and displayed in Figure 8b by an image at a scale of 1:50,000 from QGIS. The extracted areas displayed in Figure 8b have a surface of about 19,800 ha.

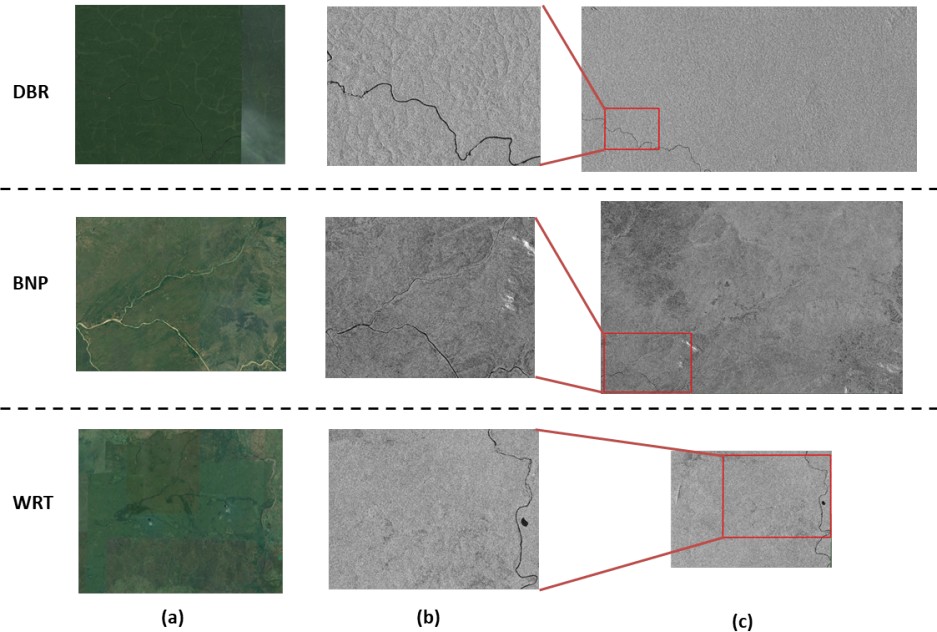

**Figure 8.** Representation of land parcels extracted from images for BNP, DBR, and WRT AOI: (**a**) RGB natural-color image of Google satellite sensor representing the extracted area; (**b**) view of the extracted area at scale 1:50,000; (**c**) image of the VH band representing the whole AOI.

## 4.3. Experiments

In this paper, a methodology to collect, preprocess and prepare SITS for use with DL algorithms is presented. The experiments resulted in three datasets that can be used for a variety of applications. To test the effectiveness of the proposed methodology and demonstrate the practicality of the datasets, three LSTM architectures suitable for prediction problems in SITS [8] are applied to the VH-band images of BNP and WRT areas. The presented application focuses on SITS prediction problem. In particular, the CNN-LSTM [42], ConvLSTM [43], and Stack-LSTM [44] models are trained with BNP and WRT SITS in order to learn how changes occur from one image to the next one, and then predict the next occurrence of an input sequence. The input size is set to five ($timestep = 5$) and the models are designed to forecast only one element ($features = 1$).

To simplify the operations, the original images are converted to JPG format with the Geospatial Data Abstraction Library (GDAL) and resized to the shape $(512, 512, 1)$. All the preprocessed images from 2015 to 2022 are used as follows: 80% for the training step and 20% for testing.

The description of the layers for the CNN-LSTM, ConvLSTM, and Stack-LSTM models are summarized, respectively, in Tables 9–11.

**Table 9.** Description of the CNN-LSTM network layers.

| Layer (Type) | Output Shape | Parameters |
|---|---|---|
| Conv2D | (None; 1; 510; 510; 8) | 80 |
| Maxpooling | (None; 1; 170; 170; 8) | 0 |
| Flatten | (None; 1; 231,200) | 0 |
| LSTM | (None; 1; 512) | 474,548,224 |
| LSTM | (None; 512) | 2,099,200 |
| Dense | (None; 262,144) | 134,479,872 |
| Reshape | (None; 512; 512) | 0 |
| Dense | (None; 512; 512) | 262,656 |

Total parameters = 611,390,032
Trainable parameters = 611,390,032
Non-trainable parameters = 0

**Table 10.** Description of the ConvLSTM network layers.

| Layer (Type) | Output Shape | Parameters |
|---|---|---|
| ConvLSTM2D | (None; None; 512; 512; 32) | 38,144 |
| Batchnormalization | (None; None; 512; 512; 32) | 128 |
| ConvLSTM2D | (None, None; 512; 512; 16) | 27,712 |
| Batchnormalization | (None; None; 512; 512; 16) | 64 |
| ConvLSTM2D | (None; 512; 512; 16) | 18,496 |
| Batchnormalization | (None; 512; 512; 16) | 64 |
| Dense | (None; 512; 512; 512) | 8704 |
| Dense | (None; 512; 512; 1) | 513 |

Total parameters = 93,825
Trainable parameters = 93,697
Non-trainable parameters = 128

**Table 11.** Description of the Stack-LSTM network layers.

| Layer (Type) | Output Shape | Parameters |
|---|---|---|
| Flatten | (None; 5; 262,144) | 0 |
| LSTM | (None; 5; 128) | 134,283,776 |
| LSTM | (None; 5; 128) | 131,584 |
| LSTM | (None; 5; 128) | 131,584 |
| Dense | (None; 262,144) | 33,816,576 |
| Reshape | (None; 512; 512) | 0 |
| Dense | (None; 512; 512) | 262,656 |

Total parameters = 168,626,176
Trainable parameters = 168,626,176
Non-trainable parameters = 0

The root mean square error (RMSE) loss function defined by (1) and the adaptive moment optimization function adam [45] are used as parameters for the compilation of the models.

$$\text{RMSE} = \sqrt{\frac{\sum_{i=1}^{n}(p_i - o_i)^2}{n}} \tag{1}$$

where $n$ represents the sample size, $p_i$ represents the predicted values and $o_i$ the observed ones.

In order to evaluate the predictive capabilities of the models, the following image quality measures are used:

- The structural similarity (SSIM) index defined by (2);
- The mean squared error (MSE) [46] defined by (3);
- The Pearson correlation coefficient (r) [47] defined by (4).

$$\text{SSIM}(x, y) = \frac{(2\mu_x\mu_y + c_1)(2\sigma_{xy} + c_2)}{(\mu_x^2 + \mu_y^2 + c_1)(\sigma_x^2 + \sigma_y^2 + c_2)} \tag{2}$$

where $x, y$ are the compared images, $\mu_x$ $\mu_y$ represent the means of $x$ and $y$, $\sigma_{xy}$ is the cross-covariance for the images, $\sigma_x^2$ and $\sigma_y^2$ represent the variances, and $c_1, c_2$ are constants used to stabilize the division when the denominator is very low [48].

$$\text{MSE} = \frac{1}{r \times c} \sum_{i=0}^{r-1} \sum_{j=0}^{c-1} (x(i,j) - y(i,j))^2 \tag{3}$$

where $x$ is the actual image, $y$ is the predicted image, and $r \times c$ is the size of images.

$$r = \frac{\sum_{i=1}^{n}(x_i - \bar{x})(y_i - \bar{y})}{\sqrt{\sum_{i=1}^{n}(x_i - \bar{x})^2}\sqrt{\sum_{i}^{n}(y_i - \bar{y})^2}} \tag{4}$$

where $x_i$ and $y_i$ represent the intensity of the $i$th pixel for the first and the second images, respectively. $\bar{x}$ and $\bar{y}$ are the mean intensity of the first and the second images, respectively, and $n$ is the sample size.

Table 12 summarizes the image quality metrics used to evaluate the DL models. One can notice that the three LSTM architectures produce satisfactory results in both AOI when the whole datasets are used. When only 50% of the dataset is used to train the models, one can notice that the predictions are less good than with more data. These results highlight the importance of having a long dataset to expect good predictive results.

**Table 12.** Evaluation metrics for CNN-LTSM, ConvLSTM, and Stack-LSTM models.

| AOI | Architecture | 100% of Dataset | | | 50% of Dataset | | |
| | | MSE | SSIM | r | MSE | SSIM | r |
|---|---|---|---|---|---|---|---|
| BNP | CNN-LSTM | 0.67 | 0.92 | 0.98 | 0.92 | 0.89 | 0.97 |
| | ConvLSTM | 0.58 | 0.95 | 0.93 | 0.76 | 0.90 | 0.87 |
| | Stack-LSTM | 0.62 | 0.90 | 0.94 | 0.68 | 0.88 | 0.91 |
| WRT | CNN-LSTM | 0.80 | 0.87 | 0.95 | 0.92 | 0.78 | 0.92 |
| | ConvLSTM | 0.93 | 0.92 | 0.90 | 0.76 | 0.90 | 0.91 |
| | Stack-LSTM | 0.78 | 0.83 | 0.92 | 0.90 | 0.78 | 0.89 |

The results of prediction with all the BNP and WRT datasets are shown in Figures 9 and 10.

### 4.4. Other Uses

The preprocessed SITS can be used for multiple applications including land use and land cover change prediction, crop monitoring, and agricultural and forestry resources analysis, to name a few. Moreover, SITS are also useful to improve the quality of the data or to analyze and understand specific features (multi-temporal filtering, temporal profile analysis, statistical operations, and so on).

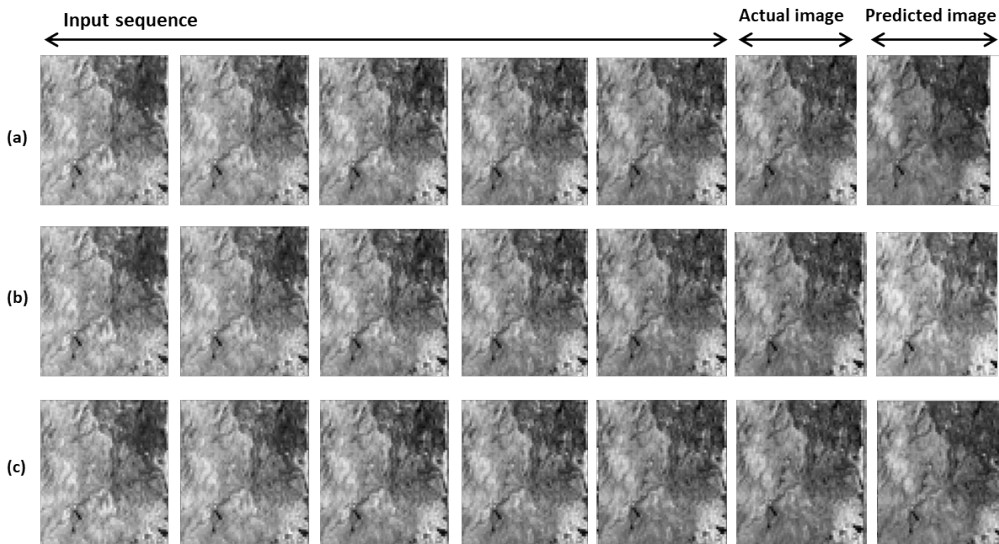

**Figure 9.** Prediction results obtained for BNP with: (**a**) CNN-LSTM model; (**b**) ConvLSTM model; (**c**) Stack-LSTM model. *timestep* = 5.

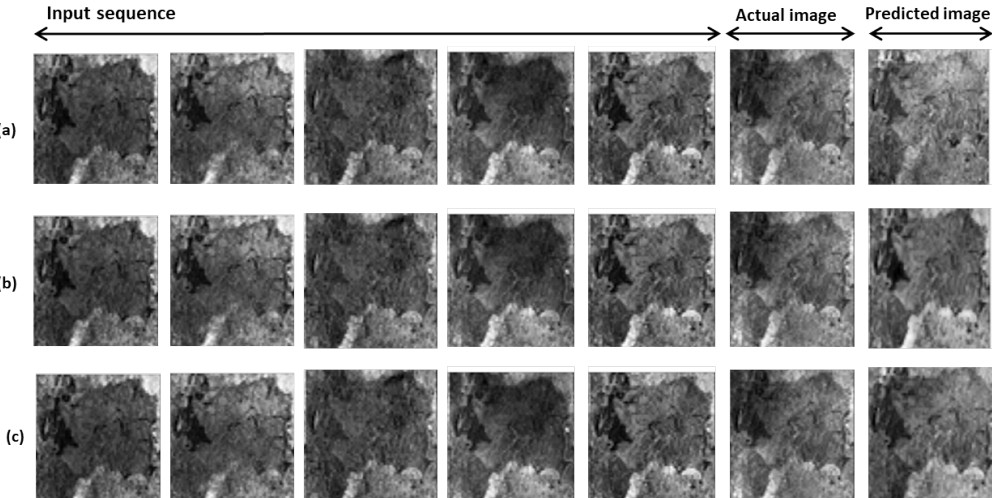

**Figure 10.** Prediction results obtained for WRT with: (**a**) CNN-LSTM model; (**b**) ConvLSTM model; (**c**) Stack-LSTM model. *timestep* = 5.

### 4.4.1. Multi-Temporal Filtering

The image time series can be used to improve the quality of S1 data by reducing the speckles (filtering operation). As shown in Figure 11, when representing filtered images one can observe that the quality of the obtained image with a multi-temporal filter is better than that obtained with filtering performed on a single product. Multi-temporal filtering not only reduces significantly the speckle ("salt and pepper" effect) but also preserves the features very well. A multi-temporal speckle filtering is only possible when a stack of time series images is available. The more images in the time series, the better the quality of the speckle filtering. Figure 11 shows the difference between the image processed with a single filter, and the image processed with a multi-temporal filter.

### 4.4.2. Temporal Profile Analysis

The analysis of the temporal profile can be important in the context of studying a particular type of crop. It allows us to understand for example the behavior of the backscatter coefficients and can be very relevant to identify the crop patterns.

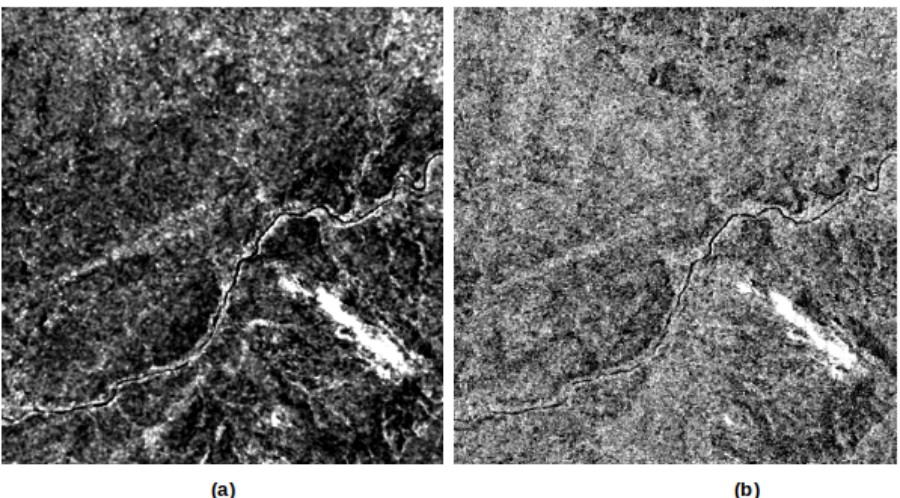

**Figure 11.** Comparison of filtering approaches on VH bands of BNP images: (**a**) filtering with a multi-temporal filter; (**b**) filtering with a single filter.

The QGIS Value tool plugin was used to identify the temporal profile of pixels over the BNP time series. For example, Figure 12 shows the variation of the value of two pixels on the time series of VH band images. The first pixel is extracted from a water surface (Figure 12a) and the second corresponds to an agricultural area.

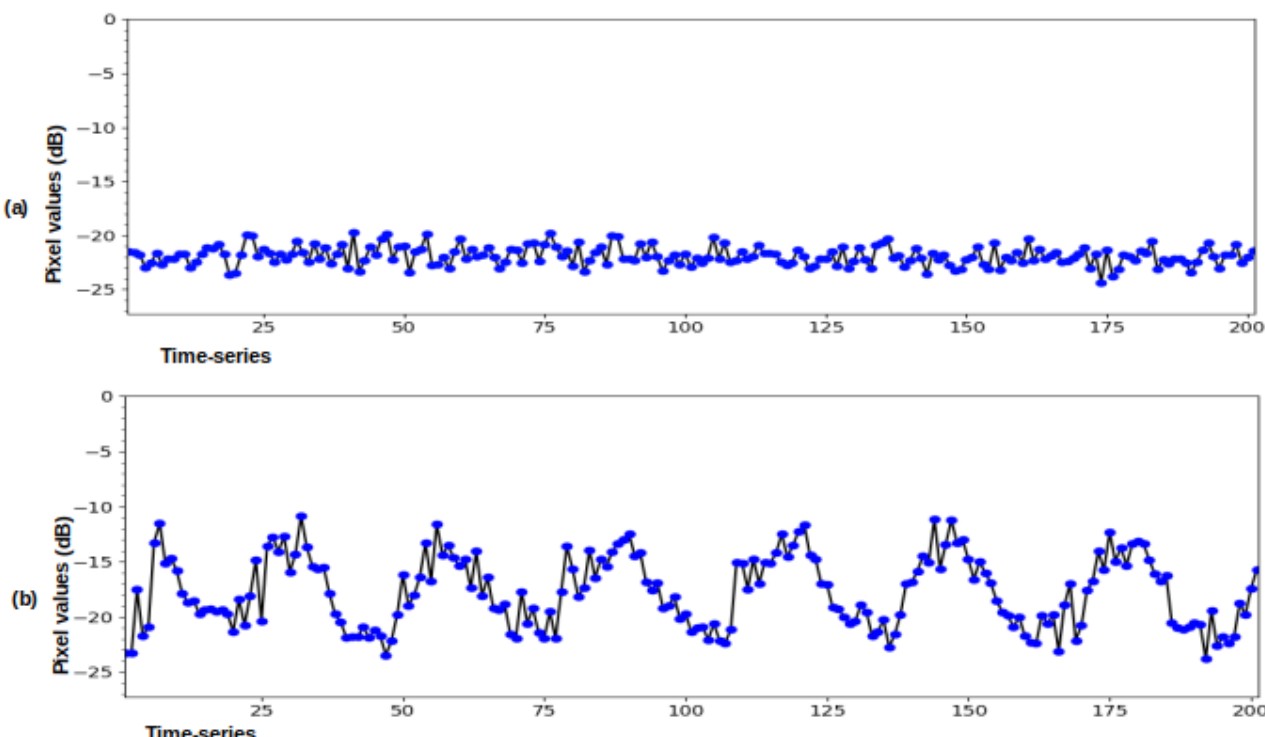

**Figure 12.** Temporal profile of pixels belonging to two soil types in the BNP image time series. The pixel values of the VH-band images are in dB and vary with the dates in the time series: (**a**) profile of the pixel with coordinates (444981, 956504) in a water surface; (**b**) profile of the pixel with coordinates (448858, 972155) in an agricultural area.

The pixel from the agricultural area has coordinates (448858, 972155) and has a particular behavior over time as shown in Figure 12b. This variation is possibly due to seasonal changes. The values increase and then decrease in a fairly regular cycle (stationary series). However, if we consider the second pixel with coordinates (444981, 956504) from another

area of the same series (water surface), the behavior presented on the curve is different, as shown in Figure 12a. In the latter case, the fluctuations in value are less important. This means that over time there is no significant change for the pixels belonging to this class.

### 4.4.3. Statistics on Time Series

In some analyses, it may be necessary to perform some calculations on the image series. In fact, it may happen that objects are not visible in some images. With the time series, it is possible to create an image consisting only of the minimum values of each pixel in the series to highlight some features. Other similar operations are also possible including mean, maximum, mode, sum, variance, and so on. Figure 13 gives an overview of a part of the BNP image consisting of the minimum values of the series. One can observe that some elements appear better compared to an image on a single date. Moreover, as with multi-temporal filtering, there is less speckle on the resulting image.

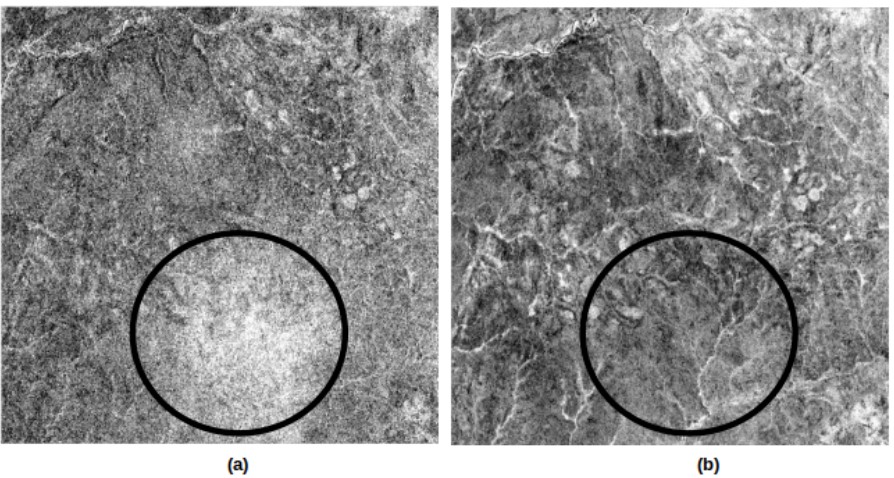

**Figure 13.** Comparison between two BNP VH-band images: (**a**) image on a single date; (**b**) image with minimumm values of pixels in the time series.

### 5. Conclusions

In this study, the researchers propose processes for automatic downloading and batch processing of S1 images. The presented processing workflow was designed based on the GPF of SNAP software. The study resulted in three time series made of preprocessed Sentinel-1A (Level-1 GRD) images in VV and VH polarizations. The researchers considered three study areas: the BNP, the DBR, and the WRT. A seven-year long-term series (1256 processed images), along with the source codes used, were made available to the public. The resulting datasets which are diverse can be used for multiple purposes, including time series prediction problems with DL algorithms such as deforestation forecasting. Moreover, the time series can help other applications such as multi-temporal speckle filtering, classification, and time series analysis for the considered areas. To validate the proposed workflow, the CNN-LSTM, ConvLSTM, and Stack-LSTM architectures were used for the next occurrence prediction in Sentinel-1 image sequences. For each of the series, missing data were found, likely due to sensor failures. Yet, having a complete series is an essential element that influences the quality of the predictions. It would, therefore, be relevant to propose algorithms to reconstruct the missing data in the Sentinel-1 image time series to have complete sequences in future works.

**Author Contributions:** Conceptualization, W.R.M., W.A., A.D. and K.; Methodology, W.R.M., W.A., A.D. and K.; Project administration, W.A. and A.D.; Software, W.R.M.; Supervision, W.A., A.D. and K.; Writing—original draft, W.R.M.; Writing—review and editing, W.R.M., W.A., A.D. and K. All authors have read and agreed to the published version of the manuscript.

**Funding:** This research study received no external funding.

**Institutional Review Board Statement:** Not applicable.

**Informed Consent Statement:** Not applicable.

**Data Availability Statement:** The source codes used for the collection and preprocessing of the images, as well as the construted datasets are available at http://w-abdou.fr/sits/ (accessed on 10 August 2022).

**Acknowledgments:** The European space agency (ESA) is thanked for providing the Sentinel-1 data used in this study and the SNAP image processing software.

**Conflicts of Interest:** The authors declare no conflict of interest.

## Abbreviations

The following abbreviations are used in this manuscript:

| | |
|---|---|
| AOF | Apply Orbit File |
| AOI | Area of Interest |
| ASF | Alaska Satellite Facility |
| BNP | Bouba Ndjida National Park |
| BNR | Border Noise Removal |
| BSB | Binational Sena Oura–Bouba Ndjida |
| CNES | National Center of Space Studies |
| CNN | Convolutional Neural Network |
| dB | Decibel |
| DBR | Dja Biosphere Reserve |
| DEM | Digital Elevation Model |
| DL | Deep Learning |
| EO | Earth Observation |
| ESA | European Space Agency |
| EW | Extra Wide Swath |
| GB | Gigabyte |
| GDAL | Geospatial Data Abstraction Library |
| GPF | Graph Processing Framework |
| GPU | Graphical Processor Unit |
| GPT | Graph Processing Tool |
| GRD | Ground Range Detected |
| h | Hours |
| ha | Hectares |
| IW | Interferometric Wide Swath |
| JAI | Java Advanced Imaging |
| LSTM | Long Short-Term Memory |
| MB | Megabyte |
| MSE | Mean Square Error |
| RMSE | Root Mean Square Error |
| S1 | Sentinel-1 |
| S1A | Sentinel-1A |
| S1B | Sentinel-1B |
| S2 | Seninel-2 |
| SITS | Satellite Image Time Series |
| SM | Stripmap |
| SNAP | Sentinel Application Platform |
| SSIM | Structural Dimilarity |

| WKT | Well-Known Text |
| WRT | Wildfire Reserve of Togodo |
| WV | Wave |
| XML | eXtensible Markup Language |

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
