# Peer review of "A Workflow for Collecting and Preprocessing Sentinel-1 Images for Time Series Prediction Suitable for Deep Learning Algorithms"

_2673-7418, doi:10.3390/geomatics2040024_

Round 1

Reviewer 1 Report

The paper deals with the development and implementation of methodologies for processing Sentinel-1 images and generating historical series for a specific Area Of Interest (AOI).

The topic is interesting, the increased availability in the recent years of multi-source satellite data at improved spatial and temporal resolution allows for the analysis and investigation of relatively long-term historical series, useful for different kinds of applications. Moreover, the access to these data is not always simple, and hence the suggestions/comments provided by the authors may help other scholars. Finally, the idea to share codes and data is well appreciated.

Said that, I have several concerns with the paper. Two of them are the most relevant:

i)                    I do not believe that the elaboration chain proposed by the author can be considered an example of deep learning algorithm. The authors produced two scripts, a python one to download Sentinel-1 images from the ASF platform, and a SNAP GPT batch to process the images. There is not any “learning” process implemented within such a processing chain or something else similar that let me think that it is not more than a well-develop elaboration chain that can be used by other researchers for building up a Sentinel-1 Level 1 VH or VV historical series for their own AOI. Therefore, I believe that author should better focus the paper on this aspect, making the necessary changes, starting for example from the title.

ii)                   It is not possible to access scripts/data suggested by the author, at the URL <<https://link/to/database/... >> This seriously impacts the scientific relevance of the paper

Others:

The used English needs to be reviewed all along the manuscript

The information provided about how to run several commands (i.e., line 170, line 179, line 201,..) can be moved to an appendix, maybe at the same level as the developed scripts.

What are the “Google satellite images”? Perhaps snapshots taken from Google Earth? 

In Figures 8-12 it must be clarified what is shown. For example, RGB natural color of… Backscatter signal in db… showing, for the SAR images, also the values scale. Geographic information should be given, or at least it has to be indicated which AOI they refer to.

In table 2, it is not clear which parameter refers to each operator.

Reviewer 2 Report

The purpose of this work is interesting, as we are facing the AI era. Some comments are as follows:

1. I don't see the differences between the proposed methodology and the current one in terms of DL.

2. Due to the use of deep learning algorithm, what aditional processing steps needs to be concerned? Currently, this manuscript hasn't it analyzed thouroughly.

3. In line 41-42, reference 11-14 include datatsets that are mostly made of Sentinel-1 images. They are radar images instead of aerial images. And they are used for target detection, like LS-SSDD-v1.0 dataset in ref 13. In terms of ship detection with AI algorithm, I don't find any benefits following the proposed methodology.

4. Currently the whole manuscript looks like a technical report, a manual of SNAP tool, I don't see the claimed contributions clearly.

Reviewer 3 Report

This paper proposed a method for S1 data collection and preprocessing for DL usage. The following questions can be considered.

1. What is the difference between traditional preprocessing and the proposed method?

2. The data is used for DL, it is appreciated for some example with DL application using the preprocessed data to validate it.

Round 2

Reviewer 1 Report

Dear authors, thanks for your efforts in considering my comments.

The main aim of the paper is now better indicated as well as the potential fields of application.

Data/scripts are now accessible, allowing scholars to use them.

My only remark is that my latest two comments have not been addressed. 

Reviewer 2 Report

Still, I don’t find this research work valuable. It may be a valued report on a technique blog instead of a research article. I don’t think the authors’ responses and modifications in the manuscript respond well to my previous concerns and questions.

1. The title and the claimed contributions are exaggerated a lot. Suppose the proposed methodology is only valuable for time-series analysis with a deep learning approach. In that case, I think the authors should narrow the scope of the tile and the claimed contributions.

Related to the added sections, two comments:

1. If this methodology is only valuable for research related to time-series analysis, what’s the difference between the proposed methodology and the current one? This vital issue is not presented well in the manuscript.

2. The difference between preprocessing steps of deep learning methods and methods without deep learning is missing in the manuscript. I suggest that the authors clarify it.

Reviewer 3 Report

This paper proposed a data collection for S1 data. The following question may help improve the manuscript:

1. Is there any common used method for time series prediction?

2. Please compare with more method in time series prediction, e.g., deep learning methods and traditional methods.

Round 3

Reviewer 2 Report

No comments.

Author Response

We are grateful to  the Reviewer for all the comments and remarks that helped to improve this work.